# Prevalence Trend and Disparities in Rheumatoid Arthritis among US Adults, 2005–2018

**DOI:** 10.3390/jcm10153289

**Published:** 2021-07-26

**Authors:** Yingke Xu, Qing Wu

**Affiliations:** 1Department of Epidemiology and Biostatistics, School of Public Health, University of Nevada Las Vegas, Las Vegas, NV 89154, USA; yingke.xu@unlv.edu; 2Nevada Institute of Personalized Medicine, College of Sciences, University of Nevada Las Vegas, Las Vegas, NV 89154, USA

**Keywords:** rheumatoid arthritis, trend, prevalence, socioeconomic status

## Abstract

Rheumatoid arthritis (RA) trends among US adults and disparities in RA patients in recent years have not been well described. We aimed to examine the trend of RA prevalence and disparities among US adults. Data from the National Health and Nutrition Examination Survey (NHANES) of the years 2005–2018 were analyzed to examine the self-reported RA prevalence trend. Age-adjusted RA prevalence stratified by race/ethnicity and socioeconomic status (SES), as well as associated linear trends, were calculated for both genders. The multivariable adjustment was used to evaluate the association between race, SES, and RA. During 2005–2018, there was no significant linear trend in the age-adjusted self-reported RA prevalence among men and women, but significant differences among people from different races, educational levels, and family poverty income ratio (PIR) groups were observed. The RA rate difference was significant for both genders and between Non-Hispanic Caucasians and Non-Hispanic African Americans (both *p*-value ≤ 0.001). Both men and women with a higher educational level and a higher PIR had a lower age-adjusted RA rate. Age-adjusted RA prevalence fluctuated for both men and women during 2005–2018. Non-Hispanic African Americans and people with low SES had significantly higher age-adjusted RA prevalence and RA risk.

## 1. Introduction

Rheumatoid arthritis (RA) is a chronic inflammatory joint disease that affects around 1% of the world population [1,2,3]. Patients with RA may develop adverse long-term outcomes such as physical and work disability, reduced quality of life, and increased mortality. Work disability is a major consequence of RA. The indirect cost of RA due to lost workability has been estimated to be nearly three times greater than treating the disease [4]. In the US, approximately 1.3 million adults suffer from RA, representing 0.6% to 1% of the adult population [3,5]. Among those afflicted with RA, the prevalence of work disability associated with RA was around 35% [6]. Moreover, the total annual health costs have been estimated to be USD 19.3 billion for the US RA patient population [3].

Several studies have examined the trend of RA prevalence in the United States, but with inconsistent results [7,8]. Park et al. found that RA prevalence among US adults decreased from 1999 to 2014 [7], while Hunter et al. suggested that the prevalence of RA in the US appeared to increase during the same period [8]. Additionally, since the two studies only assessed the RA prevalence trend before 2014, the trend after that remained unknown. Other relevant research has suggested that high socioeconomic status (SES) is significantly associated with a lower risk of developing RA [9], so a better understanding and management of SES may result in preventing the onset of RA [10]. However, studies that examined the RA prevalence trend in different SES situations have been rare in the last decade.

This study aimed to examine the recent trend of RA prevalence by gender among US adults from 2005 to 2018. We examined RA prevalence among men and women in race/ethnicity and SES groups so as to gain a more extensive understanding. We also examined the gender-specific RA risk among people in different race and SES groups after adjusting for the effects of race, SES, and related risk factors.

## 2. Materials and Methods

### 2.1. Data Source and Study Population

The National Health and Nutrition Examination Survey (NHANES) is a nationally representative survey that evaluates the health and nutrition status of adults and children in the United States. More detailed methodology and protocols have been described elsewhere [11]. In brief, the National Center for Health Statistics and the Centers for Disease Control and Prevention conduct the survey and release data in 2-year cycles. The survey utilizes a multistage probability sampling design to examine a nationally representative sample of about 5000 noninstitutionalized persons across the US. Home interviews and physical examinations are part of the data collection. During home interviews, participants are asked questions about demographic, socioeconomic, dietary, and health-related parameters. The physical examination includes medical, dental, and physiological measurements [12]. To produce reliable statistics, NHANES oversamples African Americans and Hispanic persons 60 years old and older. Sample weights in NHANES have been constructed to adjust for non-response, oversampling, and non-coverage. Our current study included adults aged 20 years or older who provided answers to questions about arthritis during the seven cycles of NHANES (2005–2006 through 2017–2018).

### 2.2. Variables

The self-reported information was used to estimate the RA prevalence in this study since several epidemiological studies have proven the validity of self-reported arthritis [13,14,15]. Participants were defined as having RA if individuals answered “rheumatoid arthritis” to the study question “Which type of arthritis was it?” Demographic variables, including age, gender, and race/ethnicity, were ascertained by questionnaire. For the race/ethnicity groups, Mexican American and Other Hispanic were merged into Hispanic, and the remaining groups were Non-Hispanic White, Non-Hispanic African American, and Non-Hispanic Other, respectively. The educational attainment and family poverty income ratio (PIR) of participants were used as SES indicators. Educational attainment was categorized into four groups: less than high school, high school graduate/ General Educational Development (GED) or equivalent, some college, and college graduate or above [16]. PIR is the ratio of family income to the family’s appropriate poverty threshold, as determined by the US Census Bureau for any given calendar year [17]. Participants were stratified into three levels based on their PIR: PIR < 1.3 (low income), 1.3 ≤ PIR < 3.5 (middle income), and ≥3.5 (high income) [18]. RA-related risk factors were considered in this study and were selected based on related literature and NHANES data availability; these factors included weight status, smoking status, and physical activity [19]. Weight and height were measured in the mobile examination center. Body mass index (BMI) was calculated as weight (kg) divided by height (m) squared and was rounded to 1 decimal place. Participants’ weight status was categorized based on BMI as follows: underweight (<18.5), normal weight (18.5–24.9), overweight (25–29.9), or obese (≥30) [20,21]. Smoking status was categorized into current smokers, former smokers, and nonsmokers [22]. Current smokers were respondents who had smoked ≥100 cigarettes during their lifetime and those who reported smoking either “every day” or “some days” at the time of the interview. Former smokers were those who reported smoking 100 cigarettes during their lifetime but currently did not smoke. Otherwise, participants were classified as nonsmokers. Physical activity was categorized as inactive and active. Participants who were sedentary or who only did basic activities, which refers to light-intensity activities such as standing and walking slowly, were considered inactive; otherwise, they were classified as active [22].

### 2.3. Statistical Analyses

Data analyses were conducted using SAS 9.4 (SAS Institute, Cary, NC, USA). Sampling weight was used to account for the complex survey design (e.g., unequal probabilities of selection) during data analyses. Age-adjusted prevalence (per 1000 population) was adjusted by the direct method to the 2000 US Census population, using 20 to 39 years, 40 to 59 years, and 60 years and older [23]. The age-adjusted prevalence of RA in every survey cycle was estimated by race/ethnicity, education level, and PIR level for each gender. Standard errors, which were employed to construct confidence intervals, were estimated using Taylor series linearization. The survey cycle was treated as a categorical variable. Orthogonal contrast was utilized to test the linear trend during the seven survey cycles in the analysis. The RA prevalence difference in race, SES, and weight status was assessed by the SAS CONTRAST program. Sex-specific logistic regression models were used to assess the associations of race, educational level, and PIR to RA while adjusting for age group, survey cycles, smoking status, weight, and physical activity. We analyzed all available data, excluding individuals with missing relevant variables. Since all variables had <10% missing data, using complete data for analyses is appropriate, as a biased estimate is unlikely. 

## 3. Results

### 3.1. Analytic Sample

From the inclusive period of 2005 to 2018, 34,171 eligible NHANES participants were used for analysis, and their weighted characteristics are presented in Table 1. The mean age of participants ranged from 46 to 48 years old. During the NHAMES study period, the proportion of Hispanics rose from 11.05% to 14.48%, whereas the percentage of Non-Hispanic Caucasians decreased from 72.13% to 64.27%. The percentage of participants with less than a high school diploma decreased from 17.28% to 10.40%, while the percentage of people with a college diploma or above increased from 26.40% to 31.13%. Meanwhile, the percentage of people with low PIR (<1.3) increased from 17.18% to 20.06%, and the percentage of participants with medium PIR (1.3–3.5) declined slightly from 37.85% to 35.92%. The distribution of RA-related risk factors for men and women is shown in Appendix A. The percentage of obesity among men and women increased during 2005–2018. Although the percentage of current smokers fluctuated in both genders, the overall pattern declined. The percentage of physical inactivity for both men and women increased during 2005–2018.

### 3.2. RA Prevalence in Both Genders

The gender-specific and age-adjusted rates of self-reported RA in 2005–2018 are shown in Figure 1. The age-adjusted RA rate among men fluctuated during 2005–2012. Afterward, the age-adjusted RA rates of men increased from 29 per 1000 population to 44 per 1000 in 2013–2018. The age-adjusted RA rate among women increased from 44 per 1000 to 50 per 1000 in the first three survey cycles; thereafter, the age-adjusted RA rate decreased to 39 per 1000 population in 2017–2018. No significant linear trend was observed for either gender in the seven survey cycles (all P_linear trend_ > 0.16).

### 3.3. RA Prevalence by Race in Men and Women

The age-adjusted rates of self-reported RA in the racial/ethnicity group for men and women are presented in Figure 2. For both genders, Non-Hispanic African Americans had a higher age-adjusted rate of RA than other groups. In men, the age-adjusted RA rate among Non-Hispanic African Americans increased from 44 per 1000 to 68 per 1000 during 2005–2010, then dropped to 44 per 1000 in 2011–2012 and increased to 58 per 1000 in the last survey cycle. For Non-Hispanic Caucasian men, the age-adjusted rate fluctuated between 23 and 39 per 1000 population from 2005 to 2018. For Hispanic men, the age-adjusted rate also fluctuated in that period, but with an overall tendency to increase. For men in each race/ethnicity group, no significant linear trend was observed in the seven cycles (all P_linear trend_ ≥ 0.09). However, the RA prevalence difference between Non-Hispanic African Americans and Non-Hispanic Caucasians was significant (*p*-value = 0.001). In women, the age-adjusted RA rate among Non-Hispanic African Americans increased from 79 per 1000 population in 2005–2006 to 91 per 1000 in 2007–2008. Then the age-adjusted RA rate decreased from 86 per 1000 population to 60 per 1000 population in 2009–2014, but the rate rose again in the last two survey cycles to a high of 83 per 1000 population. During the seven survey cycles, the age-adjusted RA rate among Non-Hispanic Caucasians fluctuated between 31 per 1000 and 48 per 1000. For Hispanic women, the age-adjusted RA rate fluctuated during 2005–2018, ranging from 51 per 1000 to 63 per 1000. No significant linear trend was observed among women in all race groups (all P_linear trend_ ≥ 0.14). When compared to Non-Hispanic Caucasian women, significant differences in RA prevalence were observed in women in the Hispanic and Non-Hispanic African American categories (both *p*-value ≤ 0.001).

### 3.4. RA Prevalence by SES in Men and Women

The pattern of the age-adjusted RA as stratified rate by education attainment in both men and women is presented in Figure 3. Compared to people with the highest education group (≥college), people from other groups had a higher age-adjusted RA rate. No significant linear trend was observed for men in each education level during the seven survey cycles (all P_linear trend_ ≥ 0.60). However, a significant difference in RA prevalence was observed between the lowest educational group (<high school) and the highest educational group (≥college) in men (*p*-value < 0.0001). No significant linear trend was observed for women in each education level group during the seven survey cycles (all P_linear trend_ ≥ 0.44). However, compared to women with the lowest educational level (<high school), women from other educational levels had significantly lower RA prevalence (all *p*-value ≤ 0.006).

The age-adjusted rate of RA by PIR for men and women is shown in Figure 4. People with the lowest family income (PIR < 1.3) had a higher age-adjusted RA rate than middle or high family incomes. Among men, no significant linear trends were observed in all PIR levels (all P_linear trend_ > 0.43). There were no significant linear trends in all three PIR groups in women (all P_linear trend_ ≥ 0.07). For both genders, the RA prevalence was significantly lower among people with 1.3 ≤ PIR < 3.5 and with PIR ≥ 3.5 when compared to people with PIR < 1.3 (all *p*-value ≤ 0.02).

### 3.5. RA Prevalence by Weight Status in Different Racial Groups, and the Association between Race, SES, and RA

The age-adjusted rates of self-reported RA by weight status in different racial groups are presented in Appendix A. No significant linear trend was observed among the four weight categories (including normal, underweight, overweight, and obese) in the Caucasian, Hispanic, and African American groups. Specifically, compared to normal people, obese people had a significantly higher age-adjusted RA prevalence in the three racial groups. Table 2 presents the adjusted association between race, SES, and RA in both genders, based on the multivariable logistic regression models. For both genders, Non-Hispanic African Americans had a significantly higher RA risk than Non-Hispanic Caucasians. Additionally, people with the highest educational level (≥college) had a significantly lower risk of RA than those with the lowest education level (<high school). Additionally, people with moderate and high family income (1.3 ≤ PIR < 3.5 and PIR ≥ 3.5) had a significantly lower risk of RA than people with low family income. 

## 4. Discussion

Although no linear trend in the age-adjusted prevalence of self-reported RA among men and women was observed during 2005–2018, there was a significant difference in RA prevalence among people from different races, educational levels, and PIR groups. Of equal significance, the association results showed that when compared to other groups, Non-Hispanic African Americans, people with an educational level of less than high school, and people with low family income (PIR < 1.3) had a significantly higher RA risk.

We found a significant difference in RA prevalence between Non-Hispanic Caucasians and Non-Hispanic African Americans in both genders. Non-Hispanic African Americans had a higher RA risk when adjusted for related risk factors. Our finding in terms of RA prevalence corresponded with Dr. Kawatkar’s study, which found that African Americans had a higher adjusted RA prevalence rate than Caucasians, Asians, and Hispanics [24]. Our findings further confirm that the disparities in income by race persist; furthermore, the data of the Census Bureau show that African Americans had the highest poverty rate among all racial groups during 2013–2019 [25]. As poverty is highly correlated with poor health outcomes and increased morbidity and mortality [26], the high poverty rates among African Americans could be one reason for their higher RA prevalence in our study. Additionally, people in poverty might suffer more stress-related RA [27]. Currently, stress is recognized as an important risk factor in the pathogenesis of autoimmune rheumatic diseases because human stress response might have an effect on the immune responses system and lead to pro-inflammatory effects and related diseases (i.e., rheumatoid arthritis) [28]. Thus, the high-level stress of people in poverty might also play a role in their higher RA prevalence. Of equal relevance, genetic factors might significantly affect this difference since RA’s heritability has been estimated to be about 60% [29]. People from various ethnicities have different human leukocyte antigen alleles and amino acid residues [30,31], leading to different RA risk factors [32]. For example, a study by Dr. Reynold et al. found that a specific amino acid is strongly associated with RA in African Americans but not in European people [33]. Therefore, underlying genetic variants could be different by ethnicities or even shared risk loci, indicating that people from varied ethnic groups might have different RA risk factors. Additionally, we found that PIR played a significant role in RA prevalence for both genders; the RA risk among people with low family income was higher than people with medium or high family income. Since lower family income is associated with lower SES, and people with low SES are more likely to smoke and have higher BMI [34,35], both are associated with a high risk of RA [10]. A recent study found that smoking may enhance oxidative stress, inflammation, and epigenetic changes, leading to RA [36]. A prior study showed that smoking is an extraneous source of oxidative stress, which might cause rheumatoid inflammation and RA [37]. Moreover, Dr. Bracke et al. reported that cigarette smoking increased the expression of matrix metalloproteinase-12 [38] and that this expression impacts the pathogenesis of RA [39]. In addition, smoking was reported to lead to extensive genome-wide DNA methylation changes [40], which play crucial roles in gene regulation and RA development [41]. Regarding high BMI, the adipose tissue of obese individuals secretes inflammatory cytokines such as leptin, tumor necrosis factor-α, interleukin-6, interleukin-1β, and monocyte chemotactic protein-1 [42], and these factors can induce an inflammatory response in individuals [43], ultimately leading to the development of RA.

In our study, Non-Hispanic African Americans and people with lower SES had higher RA prevalence and a higher risk of RA, suggesting that disparities in RA prevalence persist among minority racial/ethnic groups and disadvantaged people. Since RA can lead to pain, loss of workability, and even premature death, effective policies and strategies are needed to help these individuals in the critical areas of RA prevention, diagnosis, and treatment. Ultimately, such actions will help to reduce health disparities and improve the quality of people’s lives.

There are several limitations to this study. Due to the lack of radiographic data in continuous NHANES, RA was defined by self-reports of doctors’ diagnoses, leading to potential misclassifications; however, the Center for Disease Control recommends using self-reported, doctor-diagnosed arthritis as the case definition for estimating the prevalence of arthritis [44]. A previous study has proven the validity and reliability of self-reported data [45]. Second, data of RA’s other risk factors, such as family history, are unavailable in the NHANES, so we could not fully assess the impact of RA risk factors on prevalence.

## 5. Conclusions

In summary, the age-adjusted RA rate fluctuated during 2005–2018 in both genders. Additionally, Non-Hispanic African Americans and people with low family income (PIR < 1.3) had a significantly higher RA risk, while people with high education levels had a significantly lower RA risk. These results indicated that additional studies are needed to explore causes for these disparities in RA risk and for the development of RA prevention guidelines for people with lower SES.

## Figures and Tables

**Figure 1 jcm-10-03289-f001:**
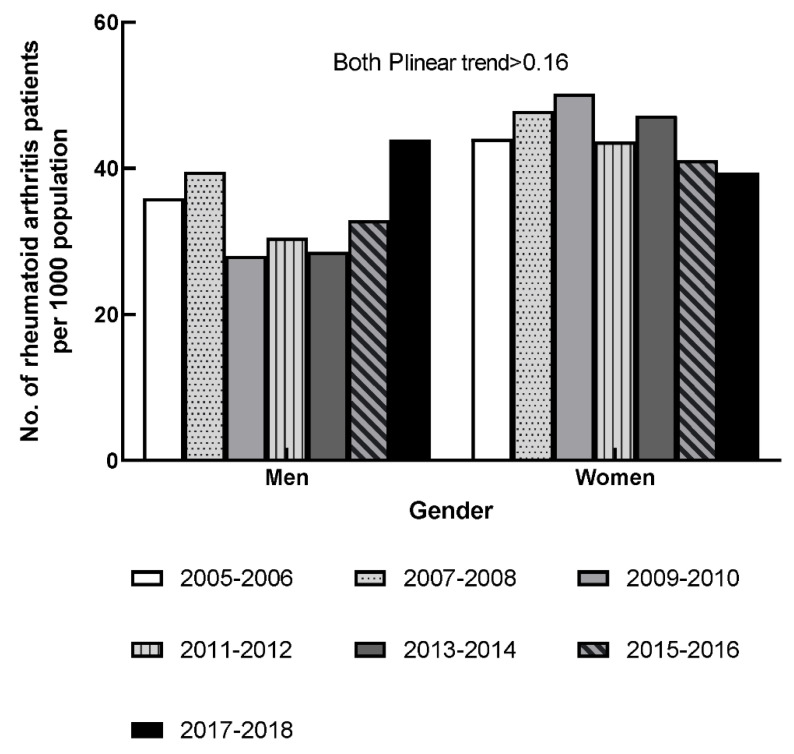
Age-adjusted prevalence of rheumatoid arthritis by gender, 2005–2006 through 2017–2018.

**Figure 2 jcm-10-03289-f002:**
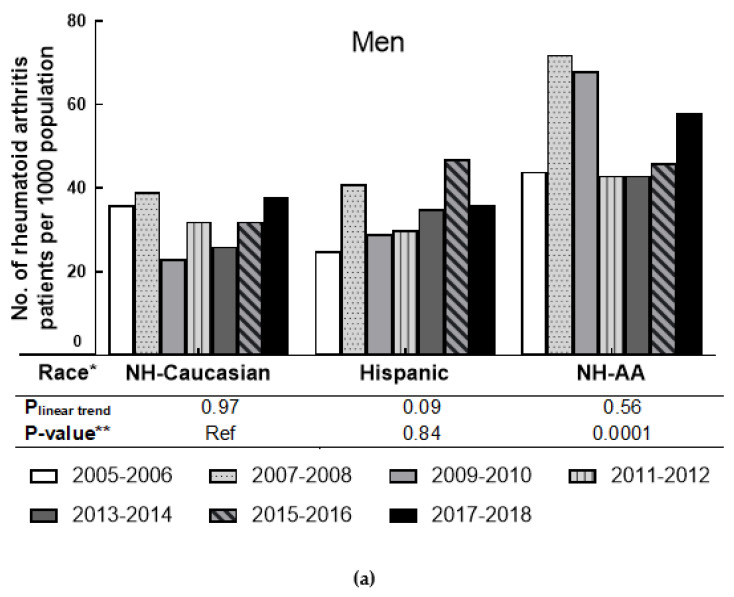
Age-adjusted prevalence of rheumatoid arthritis by race in men and women, 2005–2006 through 2017–2018. (**a**) shows the prevalence of men by race; (**b**) shows the prevalence of women by race. * includes the “other” race group not shown; ** *p*-value was based on F-test.

**Figure 3 jcm-10-03289-f003:**
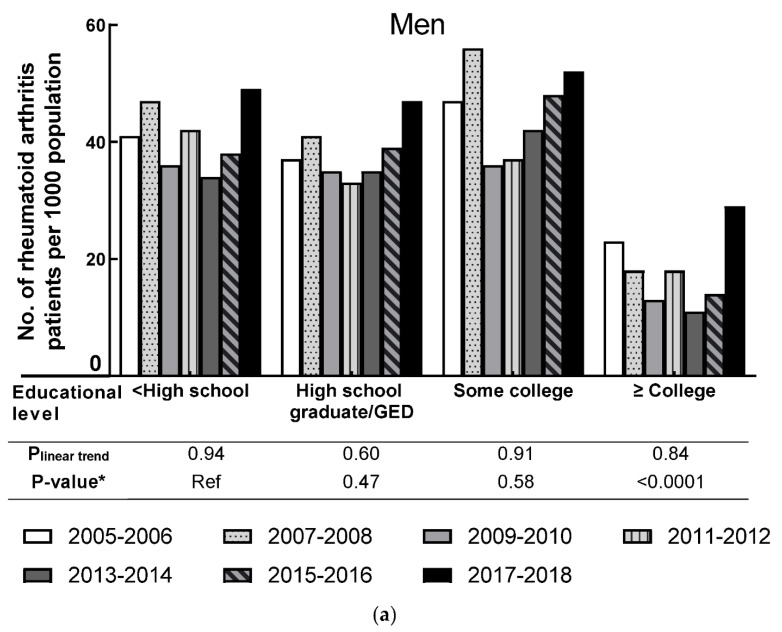
Age-adjusted prevalence of rheumatoid arthritis by education level in men and women, 2005–2006 through 2017–2018. (**a**) shows the prevalence of men by education level; (**b**) shows the prevalence of women by education level. * *p*-value was based on F-test.

**Figure 4 jcm-10-03289-f004:**
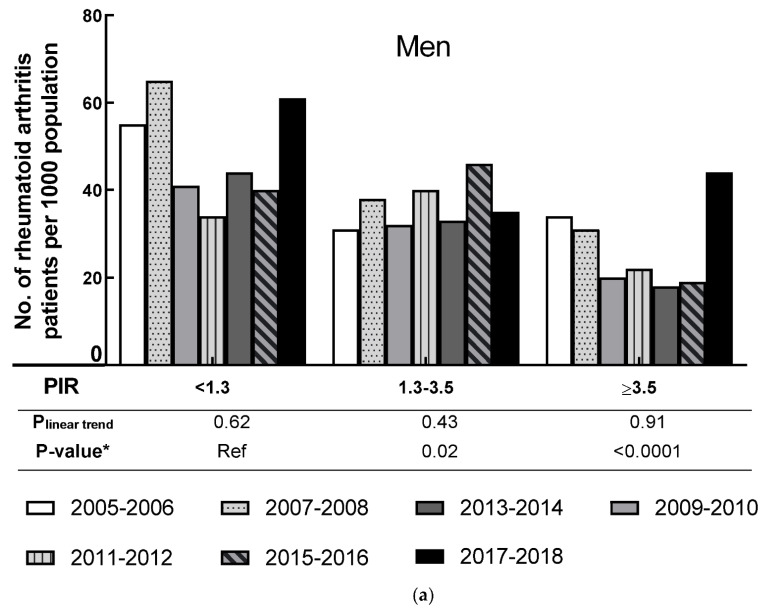
Age-adjusted prevalence of rheumatoid arthritis by poverty income ratio level in men and women, 2005–2006 through 2017–2018. (**a**) shows the prevalence of men by poverty income ratio level; (**b**) shows the prevalence of women by poverty income ratio level. * *p*-value was based on F-test.

**Table 1 jcm-10-03289-t001:** Weighted characteristics of participants in seven National Health and Nutrition Examination Surveys from 2005 to 2018.

	2005–2006 (*n* = 4459)	2007–2008 (*n* = 5084)	2009–2010 (*n* = 5399)	2011–2012 (*n* = 4801)	2013–2014 (*n* = 5094)	2015–2016 (*n* = 4846)	2017–2018 (*n* = 4488)
Age, mean (SD)	46.20 (0.75)	46.51 (0.43)	46.84 (0.51)	47.04 (0.88)	47.38 (0.38)	47.59 (0.57)	48.10 (0.65)
Women, unweighted No. (Weighted %)	2326 (51.94)	2585 (51.82)	2785 (51.96)	2442 (51.82)	2661 (51.82)	2518 (52.17)	2320 (51.77)
Race, unweighted No. (Weighted %)							
Hispanic ^a^	1012 (11.05)	1388 (12.73)	1439 (12.72)	944 (13.90)	1072 (13.89)	1447 (14.69)	954 (14.48)
NH-Caucasian	2240 (72.13)	2436 (70.23)	2691 (69.54)	1822 (67.47)	2240(66.75)	1647 (65.16)	1637 (64.27)
NH-African American	1026 (11.51)	1052 (11.04)	975 (11.11)	1243 (11.08)	1048 (11.36)	1010 (11.00)	1019 (10.83)
NH-Other	181(5.31)	208 (6.00)	294 (6.63)	792 (7.55)	734 (8.00)	742 (9.15)	878 (10.42)
Education level, unweighted No. (Weighted %)							
<high school	1205 (17.28)	1545 (20.16)	1474 (18.31)	1082 (15.80)	1049 (14.66)	1099 (13.76)	829 (10.40)
High school graduate/GED or equivalent	1065 (24.92)	1250 (25.07)	1242 (22.79)	1002 (19.87)	1141 (21.74)	1058 (20.72)	1081 (27.25)
Some college	1283 (31.40)	1318 (29.01)	1544 (30.52)	1470 (32.32)	1601 (33.12)	1459 (32.60)	1480 (31.22)
≥college	906 (26.40)	971 (25.76)	1139 (28.38)	1247 (32.01)	1303 (30.48)	1230 (32.92)	1098 (31.13)
PIR, unweighted No. (Weighted %)							
<1.3	1166 (17.18)	1551 (20.50)	1813 (21.68)	1724 (24.85)	1759 (24.79)	1567 (21.00)	1274 (20.06)
1.3–3.5	1757 (37.85)	1979 (35.01)	2023 (36.49)	1629 (34.08)	1749 (34.34)	1924 (36.78)	1855 (35.92)
≥3.5	1536 (44.97)	1554 (44.49)	1561 (41.82)	1448 (41.07)	1586 (40.87)	1355 (42.21)	1359 (44.02)

Abbreviations: NHANES = National Health and Nutrition Examination Survey; NH-Caucasian = Non-Hispanic Caucasian; NH-African American = Non-Hispanic African American; NH-Other = Non-Hispanic other; GED = General Educational Development; PIR = poverty income ratio. ^a^ Hispanic includes Mexican American and other Hispanic.

**Table 2 jcm-10-03289-t002:** Association between rheumatoid arthritis and race, education level, and poverty income ratio in both men and women: 2005–2006 through 2017–2018.

	Men *	Women *
Race		
NH-Caucasian	1 (ref)	1 (ref)
Hispanic ^a^	0.99 (0.77, 1.27)	1.41 (1.16, 1.72)
NH-African American	1.53 (1.17,1.99)	1.88 (1.58, 2.23)
Education level		
<high school	1 (ref)	1 (ref)
High school graduate/GED	0.94 (0.70, 1.25)	0.75 (0.59, 0.94)
Some college	1.14(0.84, 1.55)	0.68 (0.55, 0.83)
≥college	0.49 (0.32, 0.77)	0.40 (0.29, 0.54)
PIR		
<1.3	1 (ref)	1 (ref)
1.3–3.5	0.75 (0.59, 0.94)	0.57 (0.46, 0.71)
≥3.5	0.58 (0.44, 0.77)	0.41 (0.31, 0.53)

^a^ Hispanic includes Mexican American and other Hispanic. * adjusted for age, survey year, smoking status, weight, and physical activity.

## Data Availability

Publicly available datasets were analyzed in this study. This data can be found here: https://wwwn.cdc.gov/nchs/nhanes/Default.aspx.

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
