# Peer review of "Prevalence Trend and Disparities in Rheumatoid Arthritis among US Adults, 2005–2018"

_jcm, 2021, doi:10.3390/jcm10153289_

Round 1

Reviewer 1 Report

Well conducted work that outlines a profile of the population with RA in the US

Author Response

Thank you for the comments and suggestions. In this revision, we asked a native professional editor to review the manuscript thoroughly and revise some content to make this manuscript clearer and more concise. 

Reviewer 2 Report

Thank you for this project! Overall, important topic that is well presented and clear to the reader.

was there any discrepancy in the groups who took the survey over years between the different SES and educational level?

line 33-34: can you further specify the population , ie: adult population?

line 44: However (;)

line 79: GED ?

line 92: it'd be helpful to differentiate ex-smoker vs never smoker

line 121: was the increased Obesity noted in the RA population only or in the overall survey population?

line 254: it'll be useful to compare BMI among different race groups.

line 255: However (;)

Author Response

1) Thank you for this project! Overall, important topic that is well presented and clear to the reader.

Authors’ response: Thank you.

2) Was there any discrepancy in the groups who took the survey over years between the different SES and educational level?

Authors’ response: Thanks for your question. The percentage of participants with less than high school diploma decreased from 17.28% to 10.40%, while the percentage of people with college or above diploma increased from 26.40% to 31.13%. Meanwhile, the percentage of people with low PIR (<1.3) increased from 17.18% to 20.06%, and the percentage of participants with medium PIR (1.3-3.5) declined a little bit during 2005-2018 (Please see “3.1 Analytic sample” in the Results).

3) line 33-34: can you further specify the population , ie: adult population?

Authors’ response: Thanks for your comments. We added ‘adult’ in the revision (Please see the Introduction).

4) line 44: However (;)

Authors’ response: Thanks for your suggestions. We have double-checked related literature and consulted with the professional editor that comma should be correct in this situation.

5) line 79: GED ?

Authors’ response: Thank you for pointing this out. We spelled out the full term in the revision (Please see “2.2 Variables” in the Material and Methods).

6) line 92: it’d be helpful to differentiate ex-smoker vs never smoker

Authors’ response: Thanks for your suggestions. We re-defined the category of smoking status into three levels: never smokers, former smokers, and current smokers, and we also added related content in the manuscript (Please see “2.2 Variables” in the Material and Methods).

7) line 121: was the increased Obesity noted in the RA population only or in the overall survey population?

Authors’ response: Thanks for your question. The increased obesity rate was in the overall population instead of the RA population.

8) line 254: it’ll be useful to compare BMI among different race groups.

Authors’ response: Thanks for your suggestions. We did additional data analysis in this revision in order to determine how the RA prevalence changed by weight status in different racial groups during 2005-2018. The results show no significant linear trend among the four weight categories (including normal, underweight, overweight, and obese) in Caucasian, Hispanic, and African American. Specifically, compared to normal people, the obese people had a significantly higher age-adjusted RA prevalence in the three racial groups (Please see the Results and Supplementary figure).

9) line 255: However (;)

Authors’ response: Thanks for your suggestion. In this revision, we have double-checked related literature and consulted a professional editor, so we made revision here, a semi-colon before and a comma after ‘however” (Please see the section of Discussion).

Reviewer 3 Report

The paper by Yingke Xu and Qing Wu is an interesting portrait of the trends of rheumatoid arthritis in the US adult population.

The data was collected and analyzed thoroughly and results are clearly presented.

I do think the discussion needs to be further elaborated: on the one hand the authors discuss about the importance of genetics in the different ethnic groups, yet they do not elaborate much on the fact that ethnicity is also associated to a higher poverty risk. Also, it would be interesting for them to give more references when they stat that poorer people are more likely to smoke and to have a higher BMI. I also think it could be interesting for them to discuss the role stress might play in poorer people in the pathogenesis of the disease.

Overall the paper is very interesting, and I think that improving the discussion would make it even more valuable.

Author Response

1) The paper by Yingke Xu and Qing Wu is an interesting portrait of the trends of rheumatoid arthritis in the US adult population. The data was collected and analyzed thoroughly and results are clearly presented.

Authors’ response: Thanks for your nice comments.

2) I do think the Discussion needs to be further elaborated: on the one hand the authors discuss about the importance of genetics in the different ethnic groups, yet they do not elaborate much on the fact that ethnicity is also associated to a higher poverty risk. Also, it would be interesting for them to give more references when they stat that poorer people are more likely to smoke and to have a higher BMI. I also think it could be interesting for them to discuss the role stress might play in poorer people in the pathogenesis of the disease.

Authors’ response: We very appreciate your suggestions. In this revision, we added content about the association between race and poverty and that higher poverty might lead to poor health outcomes. In addition, we added more references in this manuscript concerning the association between lower SES and high smoking prevalence and high BMI. We also discussed the role of stress among poor people during the pathogenesis of RA (Please see the revision in the section of Discussion).